# Isothermal Nucleic Acid Amplification-Based Lateral Flow Testing for the Detection of Plant Viruses

**DOI:** 10.3390/ijms25084237

**Published:** 2024-04-11

**Authors:** Xuemei Song, Yuhao Cao, Fei Yan

**Affiliations:** 1School of Basic Medical Sciences, Health Science Center, Ningbo University, Ningbo 315211, China; caoyuhao@nbu.edu.cn; 2State Key Laboratory for Managing Biotic and Chemical Threats to the Quality and Safety of Agro-Products, Institute of Plant Virology, Ningbo University, Ningbo 315211, China; yanfei@nbu.edu.cn; 3Key Laboratory of Biotechnology in Plant Protection of MARA and Zhejiang Province, Institute of Plant Virology, Ningbo University, Ningbo 315211, China

**Keywords:** isothermal, lateral flow, detection, plant virus, on-site, LAMP, RPA

## Abstract

Isothermal nucleic acid amplification-based lateral flow testing (INAA-LFT) has emerged as a robust technique for on-site pathogen detection, providing a visible indication of pathogen nucleic acid amplification that rivals or even surpasses the sensitivity of real-time quantitative PCR. The isothermal nature of INAA-LFT ensures consistent conditions for nucleic acid amplification, establishing it as a crucial technology for rapid on-site pathogen detection. However, despite its considerable promise, the widespread application of isothermal INAA amplification-based lateral flow testing faces several challenges. This review provides an overview of the INAA-LFT procedure, highlighting its advancements in detecting plant viruses. Moreover, the review underscores the imperative of addressing the existing limitations and emphasizes ongoing research efforts dedicated to enhancing the applicability and performance of this technology in the realm of rapid on-site testing.

## 1. Introduction

Isothermal nucleic acid amplification-based lateral flow testing (INAA-LFT) has emerged as a potent technique for the detection of various pathogens in humans, animals, plants, and the environment [1,2,3]. This innovative testing method involves the amplification of genetic material, such as DNA or RNA, through isothermal amplification, followed by detection utilizing a visual readout on a lateral flow strip (LFS). Notably, INAA-LFT exhibits a considerable advantage over lateral flow immunoassay-based tests due to its heightened sensitivity [3]. The increased sensitivity can be attributed to the amplification of the pathogen’s nucleic acid within the assay [4]. 

Rapid, sensitive, and timely diagnostics are essential for protecting plants from pathogens [5]. INAA-LFT is actively being developed as a pivotal on-site rapid detection technology for pathogens [1,4]. Nevertheless, several key challenges currently impede the widespread utilization of INAA-LFT in rapid on-site detection. Here, we introduce the INAA-LFT procedure, offering a comprehensive review of its advancements in the detection of plant viruses. Furthermore, we address the pertinent factors limiting the applicability of this technology for rapid on-site testing, and also discuss ongoing efforts and proposed solutions aimed at enhancing the efficacy of INAA-LFT.

## 2. INAA-LFT Workflow

### 2.1. Amplification of the Target Nucleic Acid Fragments from Pathogens

An efficient amplification of the target nucleic acid fragments constitutes a critical foundation for the high-sensitivity detection of INAA-LFTs (Figure 1). Under isothermal conditions, nucleic acid amplification techniques, such as loop-mediated isothermal amplification (LAMP), recombinase polymerase amplification (RPA), or recombinase-mediated chain replacement nucleic acid amplification (RAA), are employed to amplify target nucleic acid fragments from pathogenic DNA or RNA [3]. They share the common goal of amplifying nucleic acids under isothermal conditions, while there are differences in their mechanisms and reaction components, which makes each technique suitable for specific detection based on factors such as target sequence complexity, sample type, and the available resources. In a recent review, Ivanov et al. provide an in-depth, systematic summary of the field diagnostic methods for plant pathogens based on different types of INAA and discussed their advantages and disadvantages [5]. In brief, LAMP has high specificity due to its multiple primer binding sites but needs a higher amplification temperature (60–65 °C). RPA and RAA do not need a high temperature and are tolerant to various sample types and inhibitors, while they have a limited multiplexing ability due to the potential primer interactions. Other INAA techniques, including helicase-dependent amplification (HDA), nickase-mediated isothermal amplification (NMA), rolling circle amplification (RCA), and cross-priming amplification (CPA), have also been developed recently [6]. They are expected to be used in LFTs.

### 2.2. Labeling of the Target Nucleic Acid Fragments from Pathogens

Various methods, including biontin, fluorescein isothiocyanate (FITC), tetramethylrhodamine (TAMRA), or digoxigenin (DIG), are utilized to label the amplified nucleic acid fragments for the detection of LFT (Figure 1). Comparative studies of different labeling methods have shown that the modification of primers within the amplification system significantly simplifies the labeling of target nucleic acid fragments, achieving the highest detection sensitivity [7]. Each label offers unique advantages and limitations. Biotin has a high binding affinity to avidin or streptavidin, allowing for highly specific detection. FITC emits strong green fluorescence upon excitation, providing sensitive and rapid detection. TAMRA produces intense red fluorescence, enhancing detection sensitivity. Digoxin provides a unique labeling option, allowing for diverse detection methods. The choice of label depends on factors such as detection sensitivity, ease of use, the availability of detection equipment, and specific assay requirements.

### 2.3. Visible Detection of Target Nucleic Acid Fragments from Pathogens

After labeling, the prepared target nucleic acid fragments are ready for visible detection using a paper-based strip. This strip contains specific zones with immobilized proteins capable of binding the labeled target nucleic acid amplification fragments. If the target nucleic acid fragments from pathogens are present in the sample, they will bind to the immobilized capture labels in the detection zone, forming a visible line that indicates a positive result [8]. A control line is also present on the strip to verify the correct functioning of the test. It captures excess labeled markers, and its appearance ensures the validity of the test (Figure 1).

## 3. Application of INAA-LFT in the Detection of Plant Viruses

### 3.1. LAMP-LFT Detection of Plant Viruses

The LAMP technique exhibits superior specificity, efficiency, and rapidity in DNA amplification under isothermal conditions, utilizing a DNA polymerase to amplify targeted DNA strands [9]. Unlike polymerase chain reaction (PCR), which necessitates cycling through various temperature steps, LAMP functions at a constant temperature, typically between 60 °C and 65 °C. This characteristic obviates the need for a thermal cycler, simplifying equipment requirements and rendering it well-suited for field applications, point-of-care diagnostics, and resource-limited settings [10,11]. 

The LAMP reaction employs four to six primers targeting multiple regions on the DNA, comprising two outer primers (forward and backward), two inner primers (forward inner and backward inner), and optionally, loop primers that expedite the reaction. In the LAMP reaction, the thermophilic enzyme Bst DNA polymerase, with strand displacement activity, synthesizes DNA at a constant temperature through a self-cycling amplification process. In this process, the displaced strand serves as a template for further amplification, leading to the formation of loop structures that contribute to the exponential amplification of the target DNA [10,11]. The endpoint of the reaction can be visually detected through turbidity, fluorescence, or a color change, depending on the chosen detection method.

Over the years, LAMP has undergone further development, integrating with other molecular approaches for plant pathogens’ diagnosis [1,2,9]. As an illustrative case, cassava brown streak virus (CBSV) and ugandan cassava brown streak virus (UCBSV) are causative agents of cassava brown streak disease in East Africa. These viruses, belonging to the genus *Ipomovirus*, family *Potyviridae*, possess a positive-sense (+), single-stranded (ss) RNA genome, leading to yield losses and the reduced marketability of cassava roots [12]. Tomlinson et al. devised primers for the rapid detection of these viruses through the reverse transcription-loop-mediated isothermal amplification (RT-LAMP). To label the target sequences, markers such as FITC, biotin, or DIG were incorporated with primers. RT-LAMP achieved amplification within 40 min, and the products were detectable using lateral flow devices containing antibodies specific to the incorporated labels [12]. 

A similar strategy was employed for the detection of tobacco rattle virus (TRV), another plant virus with an +ssRNA genome. Edgu et al. developed and optimized a mini-LAMP-lateral flow device (LFD) approach to the sensitive and specific detection of TRV in potatoes. This approach offers an economical and efficient platform for disease management in potato breeding and cultivation [13]. Notably, viral RNA purification was circumvented, and the filtered supernatant of incubation samples was diluted 1:100 with water and directly used for amplification, simplifying sample processing without the need for sophisticated laboratory equipment [13].

Recently, Lu et al. identified a novel member of the genus *Badnavirus* in the family *Caulimoviridae*, named Chinaberry tree badnavirus 1 (ChTBV1), which harbors a single molecule of non-covalently, closed, circular, double-stranded (ds) DNA in the Chinaberry tree. They developed a LAMP assay for viral detection and adapted it for the rapid visualization of results using a lateral flow dipstick chromatographic detection method [14].

In these reports, total DNA or RNA extraction from the leaves of virus-infected plants is required for detection, which increases the operation steps, the cost, and the detection time, and also has requirements for experimental equipment that are not particularly suitable for the on-site rapid testing [12,13,14].

### 3.2. RPA-LFT Detection of Plant Viruses

RPA, an additional molecular biology technique utilized for the isothermal amplification of DNA [15], shares similarities with LAMP in its capacity to facilitate DNA amplification at a constant temperature. This feature positions RPA as well-suited for field applications and point-of-care diagnostics, addressing the challenges associated with maintaining precise temperature control [16,17].

The RPA process encompasses several pivotal components, including recombinase enzymes such as recombinase A, RecA, or recombinase UvsX. These enzymes play a crucial role in facilitating strand-exchange reactions and promoting the invasion of primers into the target DNA. Additionally, the single-stranded DNA-binding protein (SSB) is indispensable for stabilizing single-stranded DNA regions, preventing reannealing. Two primers, each possessing homologous regions to the target DNA, recognize specific sequences and bind to opposite strands. The DNA polymerase, featuring strand displacement activity, extends the primers and synthesizes new DNA strands. Significantly, the reaction is isothermally conducted at a constant temperature, typically ranging between 37 °C and 42 °C.

In contrast to the LAMP-LFT amplification system, which necessitates 4~6 primers, the RPA-LFT amplification system only requires three primers for amplifying a target gene—forward and reverse primers and probes. The outer primer pair can generate specific and cloned amplification products, ensuring the accuracy of the amplification target. This streamlined primer requirement in the RPA-LFT system reduces the complexity of primer design, enhances the accuracy of detection, and lowers the detection cost compared to the more intricate primer design of the LAMP-LFT system.

RPA has diverse applications, including in molecular diagnostics, environmental monitoring, and field-based pathogen detection [6,16]. Its isothermal nature renders it suitable for resource-limited settings, and the relatively short reaction time positions it as a valuable tool for rapid DNA amplification. By incorporating specially modified probe primers into the amplification system and collaborating with LFT, RPA-LFT has emerged as a primary method for diagnosing plant viruses with varied genome types, encompassing +ssRNA, negative sense (−) ssRNA, ambisense RNA (±RNA), dsRNA, ssDNA, dsDNA, and even viroids with naked circle RNA [1,7,18,19,20,21,22,23,24,25,26,27,28,29,30] (Table 1). 

Little cherry virus 2 (LChV2), belonging to the genus *Ampelovirus* in the family *Closteroviridae* with a +ssRNA genome, causes little cherry disease (LCD) in sweet cherries (*Prunus avium*) globally. The early detection of LChV2 is crucial for controlling LCD [32]. Mekuria et al. devised an effective diagnostic method based on RPA-LFT. They developed a simple, fast, and specific RT-RPA method utilizing LChV2 coat protein-specific primers and probes, exhibiting a comparable sensitivity to RT-PCR from crude extracts. The terminally labeled amplicons were detected using a high-affinity lateral flow strip [32]. A similar approach was applied for the detection of another *Closteroviridae* family member, citrus tristeza virus (CTV), in the genus *Closterovirus* [30]. This method proved to be a powerful tool for early-stage virus detection in field samples [30,32].

Plum pox virus (PPV), a member of the genus *Potyvirus* in the family *Potyviridae* with a +ssRNA genome, causes the devastating plum pox or Sharka disease in stone fruit trees. Zhang et al. developed an efficient RPA-LFT method for PPV detection, significantly reducing the diagnostic time to as little as 20 min for the entire process, from sample preparation to results. This innovation streamlines diagnosis, facilitating both laboratory and field applications [18].

Alfalfa mosaic virus (AMV), a plant virus belonging to the genus *Alfamovirus* in the family *Bromoviridae* with an +ssRNA genome, affects a wide range of plant species worldwide. Ivanov et al. compared two methods generating labeled RPA amplicons of AMV and found that the RPA-LFT assay based on primer labeling detected 10^3^ copies of RNA in 30 min with a half-maximal binding concentration that was 22 times lower than the probe-dependent RPA-LFT. This indicates the simplicity and efficiency of labeling primers for RPA-LFT in viral diagnosis [7].

Recently, several other +ssRNA genome viruses have been targeted for detection by RPA-LFT, including cymbidium mosaic virus (CymMV) in the genus *Potexvirus* of the family *Alphaflexiviridae*, barley yellow dwarf virus (BYDV) in the genus *Polerovirus* of the family *Solemoviridae*, cowpea mild mottle virus (CPMMV) in the genus *Carlavirus* of the family *Betaflexiviridae*, actinidia chlorotic ringspot-associated virus (AcCRaV) in the genus *Emaravirus* of the family *Fimoviridae*, and bean pod mottle virus (BPMV) in the genus *Comovirus* of the family *Secoviridae* [21,22,23,26,28]. The developed RPA-LFT assay for these viruses exhibited 100 times more sensitivity than conventional reverse transcription polymerase chain reaction (RT-PCR), providing a simple, rapid, sensitive, and reliable method for viral diagnosis in the field.

Tomato spotted wilt virus (TSWV) and tomato chlorotic spot virus (TCSV), members of the genus *Orthotospovirus* in the family *Tospoviridae* with ambisense RNA genomes, cause significant yield loss in ornamental and vegetable crops worldwide. RPA-LFT assays for both viruses have been developed [27,33]. Furthermore, in the RPA reaction for TCSV, crude RNAs in the tube are incubated in the palm of the hand to generate sufficient heat for amplification. The detection limit is approximately 6 pg/µL of the total RNA from samples, providing an equipment-free, body-heat-mediated RT-RPA-LFA technique [27]. Correspondingly, methods based on RPA-LFT were developed for the detection of plant viruses with dsRNA, ssDNA, and dsDNA genomes [19,20,24,25]. 

Moreover, a multiplex assay based on RPA-LFT has also been developed to detect two or more kinds of plant viruses. Ivanov et al. successfully applied this strategy in the detection of three priority potato RNA viruses: potato virus Y (PVY), potato virus S (PVS), and potato leafroll virus (PLRV). The total assay time was 30 min. The multiplex RPA-LFT demonstrated the ability to detect at least 4 ng of PVY per gram of plant leaves, 0.04 ng/g for PVS, and 0.04 ng/g for PLRV [34]. Multiplex assays capable of simultaneously detecting multiple plant viruses provide a more comprehensive diagnostic approach, especially in regions where multiple viral pathogens may be prevalent.

In the current application of RPA-LFT in plant virus detection, an important advantage of this technique is that the crude extract from virus-infected leaves can be used for detection, which greatly simplifies the operation steps and saves time, providing it with its obvious advantage in field rapid detection [20,24,25,27,28,32,41].

### 3.3. RAA-LFT Detection of Plant Viruses

RAA shares a fundamental principle with RPA. The factor that can be used to distinguish between these two methods lies in the source of the recombinase. RPA utilizes recombinase from the T4 phage, while RAA employs recombinases from diverse sources such as bacteria and fungi. RAA has been applied to detect several viruses, showcasing its versatility in molecular diagnostics. Maize chlorotic mottle virus (MCMV) has emerged as a significant threat to maize production globally, causing maize lethal necrosis in regions of East Africa, South America, and Asia [42,43]. Duan et al. combined RAA with a clustered, regularly interspaced, short palindromic repeats and CRISPR-associated proteins 12a (CRISPR/Cas12a)-based visual nucleic acid detection system for MCMV, achieving a rapid and sensitive process completed within 45 min [36]. Expanding on this strategy, Wang et al. developed a detection system targeting sorghum mosaic virus and rice stripe mosaic virus [38]. Recently, Lei et al. integrated LFT with RAA, creating a visible system for MCMV detection [37]. Tomato brown rugose fruit virus (ToBRFV), a member of the *Tobamovirus* genus, has recently caused a pandemic in tomato and pepper production areas worldwide. Cao et al. devised an RAA-LFT for the field detection of ToBRFV with high sensitivity, demonstrating a detection limit of 2.1 × 10^1^ copies/50-μL reaction [39]. Subsequently, Zhao et al. combined RAA and CRISPR/Cas12a with LFT, enabling the simultaneous detection of four tobamoviruses—pepper mild mottle virus (PMMoV), ToBRFV, tomato mosaic virus (ToMV), and tomato mottle mosaic virus (ToMMV) [40].

RAA-LFT has the same technical advantages as RPA-LFT. Another very attractive advantage of RAA-LFT is that its cost is much lower than that of RPA-LFT, which is ideal for the batch testing of samples [44].

### 3.4. CRISPR-CAS System-Integrated LFT Detection of Plant Viruses

The CRISPR-CAS technology, renowned for its proficiency in genome editing, has found application in the INAA-LFT for plant virus detection, showing both specificity and sensitivity. In this context, CRISPR-CAS systems recognize and bind to specific amplified sequences generated by PCR, LAMP, or RPA/RAA. The CRISPR RNA (guide RNA) is meticulously designed to complement the target sequence, and the CAS protein (such as Cas12 or Cas13) undergoes activation upon binding, leading to the cleavage of the target sequence. The activation is concomitant with a detectable signal, often manifesting as a fluorescence signal. The presence of the target sequence can be identified by interpreting this signal. The integration of CRISPR-Cas with RPA confers a robust tool for nucleic acid detection, enhancing specificity through the highly precise binding of the CRISPR-Cas system to target sequences.

Marques et al. harnessed CRISPR-Cas12a and CRISPR-Cas13a/d systems to detect the viral DNA amplicons generated by PCR or isothermal amplification, focusing on three RNA viruses: tobacco mosaic virus (TMV), tobacco etch virus (TEV), and potato virus X (PVX). They innovatively adapted the detection system to circumvent the costly RNA purification step and achieve a visible readout through lateral flow strips, enabling rapid viral diagnostics within a timeframe of half an hour [35]. Addressing key viruses in rice, rice stripe virus (RSV), and rice black-streaked dwarf virus (RBSDV), Zhu et al. devised a CRISPR/Cas12a-assisted LAMP-LFT system to detect these viruses, along with the bacterial pathogen *Xanthomonas oryzae* pv. *oryzae* (Xoo). The heightened sensitivity of this system reached as low as nine or three copies [31]. In the LAMP-LFT assay for MCMV detection, assisted by CRISPR-Cas12a, the detection limit achieved an impressive low of 2.5 copies of the coat protein (CP) gene of MCMV [37]. 

The above results indicate that that the assistance of CRISPR-CAS in the detection system improves the detection limit to a very low level. Meanwhile, it should be noted that the specific implementation of CRISPR-CAS in detection may vary based on the target application and the desired detection method. The continued exploration and optimization of these technologies are required for a variety of diagnostic and research purposes.

## 4. Factors Influencing the Implementation of INAA-LFT for the On-Site Detection of Plant Viruses

The utilization of INAA-LFT for the on-site detection of plant viruses is intricately influenced by several key factors, including sensitivity and specificity, rapid testing duration, portability and field-friendly device characteristics, and cost-effectiveness. These factors play a pivotal role in shaping the feasibility, effectiveness, and practicality of implementing INAA-LFT in field settings. 

### 4.1. Sensitivity and Specificity

Detection sensitivity stands as a pivotal metric when assessing a particular detection technology. Within the established INAA-LFT systems, detection limits typically fall within the range of 2.5–20 viral copies per reaction, often surpassing or reaching the detection threshold of real-time quantitative PCR. This level of sensitivity adequately caters to the requirements of field detection [26,28,31,37]. Notably, when diagnosing a plant sample exhibiting virus symptoms, the sample generally contains a sufficient viral load for detection by INAA-LFT, thus mitigating the necessity for an exceptionally high detection sensitivity. However, in scenarios involving the early detection of samples, where virus symptoms may not be apparent and the goal is to ascertain the presence of a specific virus, a heightened detection sensitivity becomes imperative. 

The accurate detection of viral nucleic acids using INAA-LFT is pivotal, and a higher specificity is instrumental in mitigating false positives, particularly in samples infected with multiple viruses. This necessitates stringent requirements for primer design during the nucleic acid amplification process, a critical consideration not only for INAA-LFT but also for other nucleic-acid-based detection technologies. Furthermore, in the context of multiplex assays aiming to detect two or more distinct viruses, even within the same genus, the significance and complexity of the primer design are heightened—a formidable challenge for researchers [34,40].

### 4.2. Detection Duration

The rapid generation of results is pivotal for on-site detection. The speed of INAA-LFT significantly influences the ability to provide timely confirmations in field settings. The current testing system typically completes the entire process, from sample collection to the presentation of test results, within one hour. The fastest recorded duration is 20 min, while the slowest is about 50 min [19,26,30,31]. Although there is the potential to optimize the system framework to achieve a testing duration of 20 min, this timeframe still falls short of meeting the demand for the rapid on-site detection of plant viruses [19,26,30].

### 4.3. Ease of Operation

In the reaction process of INAA-LFT, many components are required to exert biochemical activity to achieve the amplification of nucleic acids, the generation of signals, and the visualization of signals. Under laboratory conditions, these components usually participate in the reaction in order to ensure the stability of the reaction system and avoid the occurrence of various false results. In the INAA-LFT’s currently established detection system for plant viruses, scientific researchers have tried their best to mix various ingredients into a tube (all in one) and simplify the operating steps as much as possible without affecting the activity of each ingredient. For example, reverse transcription is combined into an amplification reaction and nucleic acid amplification is combined with CRISPR/Cas12a in one mixture.

In the intricate reaction process of INAA-LFT, various components must exhibit biochemical activity to achieve nucleic acid amplification, signal generation, and signal visualization. In laboratory conditions, these components are meticulously curated to ensure reaction system stability and mitigate the risk of false results. In the currently established INAA-LFT detection system for plant viruses, researchers have diligently worked to consolidating multiple components into a single reaction tube (all in one) and simplify operational steps without compromising the efficacy of each component. Notably, processes such as reverse transcription are amalgamated into the amplification reaction, and nucleic acid amplification is seamlessly integrated with CRISPR/Cas12a within a single mixture [28,37,39,40]. Moreover, the development of a nucleic acid extraction-free process that is seamlessly incorporated into the all-in-one reaction further streamlines the INAA-LFT steps, enhancing simplicity [25,28,32,37,40].

However, based on our practical experience, in actual field operations, operators are still required to utilize pipettes for sample addition, buffer incorporation, and subsequent product dilution. This necessity raises the basic technical threshold for operators and concurrently poses limitations on the broader application of INAA-LFT in field detection.

## 5. The Future Trajectory of INAA-LFT in Detection of Plant Virus

### 5.1. Expedited Detecting Duration

In the existing literature on the detection of plant viruses using INAA-LFT, the detection time typically spans from 20 to 50 min, posing a limitation for widespread on-site applications. The primary time is allocated to nucleic acid amplification, which is determined by the performance of the recombinase in the system. Nucleic acid amplification usually takes around 15 min to reach a detectable level. If this time can be reduced to less than 8–10 min, it would significantly advance the practical use of this detection technology in the field (Figure 2). Despite the successful integration of reverse transcription, nucleic acid amplification, and signal production into one tube, challenges remain regarding the need to simplify or omit operational steps. The quest for novel recombinases with higher activity holds promise for achieving a faster INAA-LFT detection. Additionally, exploring alternative amplification strategies or optimizing reaction conditions may reduce the necessary time.

### 5.2. Operational Simplification and Automation

The envisioned scenario involves field personnel without specialized molecular technique training, allowing for the effortless conduction of INAA-LFT procedures and interpretation of the results. At present, several pipetting steps are essential, necessitating further optimization. The developmental focus on simplifying the system, enhancing user-friendliness, and the potential automation of operations is a promising direction. Integrated systems managing sample preparation, testing, and the analysis of results could revolutionize the landscape. Moreover, coupling INAA-LFT with advanced data management systems has the potential to facilitate result collection, storage, and analysis, contributing to the establishment of a comprehensive database for monitoring and managing plant virus outbreaks (Figure 2). Additionally, exploring technologies such as robotics or microfluidics could further enhance automation and ease of use.

### 5.3. Portable Devices

On-site INAA-LFT detection requires specific instruments, and the absence of a commercially available all-in-one machine poses a current limitation. The ideal INAA-LFT device should be portable, tailored for field use, and resistant to the environmental conditions prevalent in agricultural settings. Considerations include size, weight, durability, testing throughput, and resilience against varying environmental conditions. Recent innovations, such as our development of a portable toolbox powered by a car cigarette lighter, showcase progress in addressing portability concerns. However, the existing system still involves several pipetting steps, highlighting the need for further streamlining. Anticipating all-in-one machines that integrate sample handling, automated pipetting, the interpretation of results, GPS positioning, and multi-environment adaptability is a pivotal advancement that is eagerly awaited in the field (Figure 2). Moreover, the integration of advanced energy sources, such as solar power, could play a transformative role in enhancing the sustainability and portability of these devices, making them more practical and efficient for on-site detection applications.

### 5.4. Regulatory Approvals and Standardization

Regulatory considerations and adherence to standards play a pivotal role in shaping the future trajectory of a technology. At present, fluorescence quantitative real-time polymerase chain reaction (qPCR) remains the gold standard for the sensitive and specific detection and quantification of pathogenic nucleic acids. In contrast, INAA-LFT-based assays lack standardized protocols, relying on individual laboratory conditions and procedures. A fundamental reason for this disparity is the absence of a standardized instrument. Establishing a unified protocol for INAA-LFT, grounded in standard instruments, is imperative to ensure consistency and reliability across different tests and platforms. Collaborative efforts between researchers, regulatory bodies, and industry stakeholders are essential to define and implement standardized procedures (Figure 2). Additionally, emphasizing the importance of validation studies and external quality-control measures will bolster the credibility and acceptance of INAA-LFT in the scientific community and regulatory frameworks.

### 5.5. Integration of Artificial Intelligence (AI) for Data Analysis

With the continuous advancement of technology, the integration of Artificial Intelligence (AI) in the colorimetric data analysis of LFT can further enhance the efficiency and accuracy of results’ interpretation in the context of INAA-LFT. For instance, harnessing the AI algorithms’ ability to swiftly process large datasets, AI can identify colorimetric patterns in LFT and, based on the standard colorimetric chart used for the absolute quantification of a pathogen in LFT, estimate the copy number of the pathogen in actual samples (Figure 2). This approach was investigated via the sensitive and quantitative detection of a COVID-19-neutralizing antibody by LFT [45,46]. Exploring the potential synergies between INAA-LFT and AI will open new avenues to improve the overall performance and applicability of this technology in diverse settings.

## Figures and Tables

**Figure 1 ijms-25-04237-f001:**
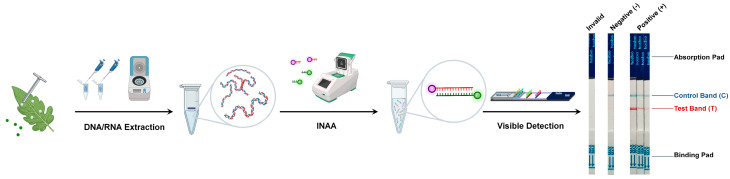
Workflow of the INAA-LFT. Total DNA or RNA is extracted from the leaves of virus-infected plants, and then used as template for INAA. The labeled target nucleic acid fragments are visualized on a paper-based strip. A control line is also present on the strip to verify the correct functioning of the test.

**Figure 2 ijms-25-04237-f002:**
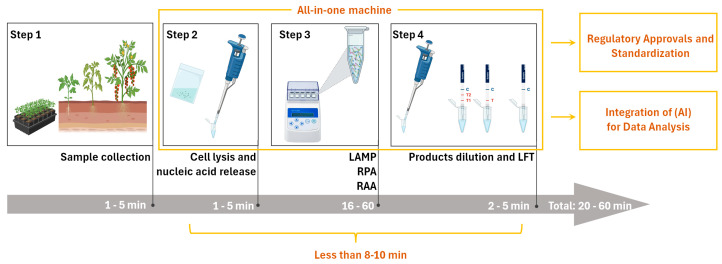
Optimized workflow of INAA-LFT at present, and its future trajectory in the detection of plant virus. In the current INAA-LFT system, starting with sampling, there are three subsequent processes, including reaction preparation, isothermal amplification, and product dilution and detection, which take a total of 20–60 min. In the future, if the detection time can be reduced to 8–10 min, sample processing, reaction, and detection can be completed using one device, and can be combined with artificial intelligence (AI) to achieve standardization (parts framed by yellow lines). This will open new avenues for improving the overall performance and applicability of this technology in diverse settings.

**Table 1 ijms-25-04237-t001:** Plant viruses detected through INAA-LFT.

Type of Amplification	Virus	Type of Viral Genome	Testing Duration	Sensitivity *	Ref.
LAMP	Cassava brown streak virus	+ssRNA	40 min	2.9 ng total RNA/μL	[12]
	Ugandan cassava brown streak virus				
	Tobacco rattle virus	+ssRNA	<50 min	78 pg template/μL RNA	[13]
	Chinaberry tree badnavirus 1	dsRNA	45 min	0.5 pg/reaction	[14]
	Rice stripe virus	±RNA	50 min	3 copies per reaction	[31]
	Rice black-streaked dwarf virus	dsRNA	50 min		
RPA	Alfalfa mosaic virus	+ssRNA	30 min	10^3^ copies of RNA in reaction	[7]
	Plum pox virus	+ssRNA	20 min	1.0 fg transcripts/reaction	[18]
	Rice black-streaked dwarf virus	dsRNA	20 min	Similar to RT-PCR	[19]
	Milk vetch dwarf virus	ssDNA	30 min	10^1^ copies per reaction	[20]
	Cymbidium mosaic virus	+ssRNA	30 min	-	[21]
	Barley yellow dwarf virus	+ssRNA	20 min	100 pg/μL	[22]
	Bean pod mottle virus	+ssRNA	<90 min	500 pg/μL	[23]
	Tomato yellow leaf curl virus	ssDNA	30 min	0.5 pg DNA per reaction	[24]
	Piper yellow mottle virus	dsDNA	30 min	10 times more sensitive than PCR	[25]
	Tomato chlorotic spot virus	±RNA	15 min	6 pg/μL of total RNA	[27]
	Actinidia chlorotic ringspot-associated virus	+ssRNA	<40 min	20 viral copies	[28]
	Citrus tristeza virus	+ssRNA	15–20 min	141 fg of RNA when cDNA used as a template	[30]
	Little cherry virus 2	+ssRNA	-	Similar to RT-PCR	[32]
	Tomato spotted wilt virus	±RNA	15 min	10 fg TSWV CP transcripts	[33]
	Potato virus Y (PVY)	+ssRNA	30 min	4 ng of PVY per g of plant leaves	[34]
	Potato virus S (PVS)	+ssRNA	30 min	0.04 ng of PVS per g of plant leaves	[34]
	Potato leafroll virus (PLRV)	+ssRNA	30 min	0.04 ng of PVS per g of plant leaves	[34]
	Tobacco mosaic virus	+ssRNA	40 min	-	[35]
	Tobacco etch virus	+ssRNA	40 min	-	[35]
	Potato virus X	+ssRNA	40 min	-	[35]
RAA	Maize chlorotic mottle virus	+ssRNA	45 min	0.02 ng of total RNA	[36,37]
	Sorghum mosaic virus	+ssRNA	30 min	10^7^ dilution	[38]
	Rice stripe mosaic virus	-ssRNA	30 min	10^7^ dilution	[38]
	Tomato brown rugose fruit virus	+ssRNA	20 min	10^1^ copies /reaction	[39]
	Pepper mild mottle virus	+ssRNA	<1 h	-	[40]
	Tomato mosaic virus				[40]
	Tomato mottle mosaic virus				[40]

*: the data are for reference only since they are obtained directly from the literature, where different references (total DNA/RNA, viral DNA/RNA, dilution fold, etc.) are used to analyze sensitivity.

## Data Availability

No new data were created in this work.

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
