# Peer review of "Isothermal Nucleic Acid Amplification-Based Lateral Flow Testing for the Detection of Plant Viruses"

_ijms, 2024, doi:10.3390/ijms25084237_

Round 1

Reviewer 1 Report

Comments and Suggestions for Authors

The manuscript gives useful characterization of a perspective direction in analytical developments. The initial literature data are carefully collected. Unfortunately, I don't think that the level of the analysis that was given for state-of-the art and perspectives accords to the level of the International Journal of Molecular Sciences.

The main shortcomings of the prepared manuscript are as follows:

1. Significant part of the review's volume takes simple extended repetitions of the general principles of common experimental approaches (processes occurring during various isothermal amplifications, compounds of lateral flow test strips, etc.) without assessing the specifics of their use in the control of viral pathogens. Such material's presentation is more typical for educational reviews.

2. The descriptions of the examples in the Sections 3.1-3.3 present occasional details of different developments with limited possibility of their comparison and integral discussion. The Sections need in more focused characterization of practically demonstrated advantages and limitations of the three considered amplification techniques in their application to plant viruses and integration with LFTs.

3. The Section 4 should give not only simple statement of the existing limitations, but also clearly visible recommendations to overcome them. Unfortunately, the Section 5 is not helpful for this, as well as it describes some ideal testings without concrete practical recommendations about changes in the existing protocols.

4. The review about plant viruses should include detailed characterization of the initial actions with plant samples. In what situations is plant testing carried out? What parts of plants should be taken to find different viruses? What sample preparation procedures are used? All these issues are not specified in the review.

5. Main labels that are used LFTs are indicated, but without their comparative analysis, existing comparisons, reasons to choose one or other candidate for different purposes.

6. The necessity of selective assays is reasonably stated, but without consideration of specific situations with different pathogens. How the existing assays cover variety of the pathogens' strains and how this coverage should be improved without false-positive results for related but less dangerous variants of the pathogens?

7. The comments about AI applications are given on the base of common knowledge, without consideration of any real (published) activities in this field.

8. The place of the work among predecessors is not characterized. The aim of the review is very similar with the earlier published paper «The potential use of isothermal amplification assays for in-field diagnostics of plant pathogens» (Plants. 2021; 10 (11): 2424. doi: 10.3390/plants10112424). The addressing to it and clarification of novel specific tasks will be reasonable. Another earlier review «Advanced DNA-based point-of-care diagnostic methods for plant diseases detection» (Front Plant 2017, 8: 2016. doi: 10.3389/fpls.2017.02016) was formally cited as ref. [17], but without its meaningful consideration.

9. Table 1. The collected testing durations are given without specification do they include sample preparation or not. Such key assay's characteristic as limits of detection are indicated only for some examples with their extended consideration in the review, but do not checked for examples collected in the review. This does not allow giving a comparative assessment for the capabilities of different methods.

Some additional minor remarks reflecting low quality of data consideration:

10. Lines 35-37. The actions after pathogen's detection for human and for plants are very different. So the given addressing to COVID-19 is not a good clarification for the actual need in rapid testing of plant diseases.

11. Lines 46-49. INAA includes more variants, and the listed LAMP, RPA and RAA are only the most popular ones to date.

12. Lines 54-55. Biotin is not a label for visual detection. It is used as an intermediate compound in the formation of the detected complexes in LFTs.

Author Response

The manuscript gives useful characterization of a perspective direction in analytical developments. The initial literature data are carefully collected. Unfortunately, I don't think that the level of the analysis that was given for state-of-the art and perspectives accords to the level of the International Journal of Molecular Sciences.

 The main shortcomings of the prepared manuscript are as follows:

  1. Significant part of the review's volume takes simple extended repetitions of the general principles of common experimental approaches (processes occurring during various isothermal amplifications, compounds of lateral flow test strips, etc.) without assessing the specifics of their use in the control of viral pathogens. Such material's presentation is more typical for educational reviews.

Reply: Thank you for very professional comments. At the beginning of this review, we gave a brief overview of the LFT process, which is helpful for beginners to this technique. The application of these techniques to different virus detection is described in section 3.1-3.3 in detail.

  1. The descriptions of the examples in the Sections 3.1-3.3 present occasional details of different developments with limited possibility of their comparison and integral discussion. The Sections need in more focused characterization of practically demonstrated advantages and limitations of the three considered amplification techniques in their application to plant viruses and integration with LFTs.

Reply: Thank you for your comment. We've now added practical features of LAMP, RPA, and RAA technologies to the corresponding sections (Lines 50-62, Lines134-137, Lines223-226, Lines247-249).

  1. The Section 4 should give not only simple statement of the existing limitations, but also clearly visible recommendations to overcome them. Unfortunately, the Section 5 is not helpful for this, as well as it describes some ideal testings without concrete practical recommendations about changes in the existing protocols.

Reply: In section 4, we try to analyze factors that affect the application of these technologies in the field, based on our personal understanding, but not directly point out the limitations of current technologies. In addition, for certain factors, such as the detection duration, we also provide recommendations in Section 5 on how to save more time on the test (Lines351-353).

  1. The review about plant viruses should include detailed characterization of the initial actions with plant samples. In what situations is plant testing carried out? What parts of plants should be taken to find different viruses? What sample preparation procedures are used? All these issues are not specified in the review.

Reply: For detection of plant viruses, leaves of the infected plants could be used for detection. We have now added this information in section 3.1-3.3 (Lines134-137, Lines223-226, Lines247-249).

  1. Main labels that are used LFTs are indicated, but without their comparative analysis, existing comparisons, reasons to choose one or other candidate for different purposes.

Reply: Thanks for the comment. We have now added this information to show the advantage of each label (Lines70-76).

  1. The necessity of selective assays is reasonably stated, but without consideration of specific situations with different pathogens. How the existing assays cover variety of the pathogens' strains and how this coverage should be improved without false-positive results for related but less dangerous variants of the pathogens?

Reply: Thank you for your comment. For viral detection, primers designed for INAA guarantee specificity. Isolates of viruses (like strains of bacterial or fungal pathogen), have unique genomes. In experiments, specificity should be carefully investigated by analyzing multiple controls to avoid false positives or false negatives. If we want to expand the detection range, for example to detect all viruses of a certain genus, we need to design primers based on conserved sequences. In this case, multiple controls are also necessary.

  1. The comments about AI applications are given on the base of common knowledge, without consideration of any real (published) activities in this field.

Reply: Thank you for your comment. AI is rapidly penetrating various fields at an alarming rate. Actually, it has been used in the detection of COVID-19 antibody by LFT (we have now cited two references [45, 46] in the text, Lines415-416). We believe it will soon enter the field of plant virus detection. We cover this briefly in the last paragraph, which should be somewhat instructive.

  1. The place of the work among predecessors is not characterized. The aim of the review is very similar with the earlier published paper «The potential use of isothermal amplification assays for in-field diagnostics of plant pathogens» (Plants. 2021; 10 (11): 2424. doi: 10.3390/plants10112424). The addressing to it and clarification of novel specific tasks will be reasonable. Another earlier review «Advanced DNA-based point-of-care diagnostic methods for plant diseases detection» (Front Plant 2017, 8: 2016. doi: 10.3389/fpls.2017.02016) was formally cited as ref. [17], but without its meaningful consideration.

Reply: Thank you for your information. We missed this reference (Plants. 2021) and have now cited it in the text (Line35, Lines54-56). In addition, we added more meaningful consideration for citing the Front Plant paper (Lines60-62).

  1. Table 1. The collected testing durations are given without specification do they include sample preparation or not. Such key assay's characteristic as limits of detection are indicated only for some examples with their extended consideration in the review, but do not checked for examples collected in the review. This does not allow giving a comparative assessment for the capabilities of different methods.

Reply: Thank you for your comment. We have now added information on sample preparation and nucleic acid extraction in section 3.1-3.3 (Lines134-137, Lines223-226, Lines247-249). Also, the message that nucleic acid extraction is not required in the developed RPA and RAA assay was indicated in section 3.2 and 3.3.

Some additional minor remarks reflecting low quality of data consideration:

  1. Lines 35-37. The actions after pathogen's detection for human and for plants are very different. So the given addressing to COVID-19 is not a good clarification for the actual need in rapid testing of plant diseases.

Reply: We deleted this sentence and rewrote it.

  1. Lines 46-49. INAA includes more variants, and the listed LAMP, RPA and RAA are only the most popular ones to date.

Reply: We added other variants, including HAD, NMA, RCA, etc (Line35).

  1. Lines 54-55. Biotin is not a label for visual detection. It is used as an intermediate compound in the formation of the detected complexes in LFTs.

 Reply: Corrected (Lines65-66).

Reviewer 2 Report

Comments and Suggestions for Authors

This review, entitled "Isothermal Nucleic Acid Amplification-based lateral flow testing for detection of plant viruses," provides a summary of the techniques used in detecting plant viruses. The authors' effort in optimizing the time and portability of these techniques is appreciated. The article is well-written and makes extensive and innovative references.

I suggest including figures to provide a graphical overview of the techniques and real pictures to illustrate how both positive and negative tests appear to the reader. Additionally, it would be beneficial to include the sensitivity of these techniques in Table 1. Providing an estimated price per test could also be useful for potential users.

The inclusion of more data and a figure summarizing the described techniques in the text would greatly enhance reader interest and make it a valuable reference for those seeking to familiarize themselves with these detection techniques.

I noticed a couple of errors:

  • Line 164: "primer labeling detected 103 copies of RNA" – Is it 103 or 103?
  • Line 211: "2.1 × 101 copies/50-μL reaction" – Is it 101 or 101? Please verify these potential numerical mistakes.

In conclusion, I believe the article can be published, but I recommend implementing these improvements.

Author Response

We appreciate the reviewer for the positive comments on the manuscript and useful suggestion to improve it. We have now revised the manuscript accordingly and respond to comments one by one.

This review, entitled "Isothermal Nucleic Acid Amplification-based lateral flow testing for detection of plant viruses," provides a summary of the techniques used in detecting plant viruses. The authors' effort in optimizing the time and portability of these techniques is appreciated. The article is well-written and makes extensive and innovative references.

Reply:Thank you for your comments. We accepted all of them and revised the manuscript accordingly.

I suggest including figures to provide a graphical overview of the techniques and real pictures to illustrate how both positive and negative tests appear to the reader. Additionally, it would be beneficial to include the sensitivity of these techniques in Table 1. Providing an estimated price per test could also be useful for potential users.

Reply:Agreed. We have now added a figure that outlines the technique and shows how the results of positive and negative samples are visualized (Figure 1). Also, we added the sensitivity reported in the references into Table 1.

The inclusion of more data and a figure summarizing the described techniques in the text would greatly enhance reader interest and make it a valuable reference for those seeking to familiarize themselves with these detection techniques.

Reply:Thank you for your constructive comments. We have now added a figure summarizing the technique (Figure 1).

I noticed a couple of errors:

Line 164: "primer labeling detected 103 copies of RNA" – Is it 103 or 103?

Reply:This should be a scientific count (103). Corrected.

Line 211: "2.1 × 101 copies/50-μL reaction" – Is it 101 or 101? Please verify these potential numerical mistakes.

Reply:This should be a scientific count (101). Corrected.

In conclusion, I believe the article can be published, but I recommend implementing these improvements

Reply:Thank you for your constructive comments, which greatly improved the manuscript.

Reviewer 3 Report

Comments and Suggestions for Authors

In this paper, the INAA-LFT process has been very well described in detail.

This review addresses the limitations of new technology and suggest the enhancing efficacy detect plant virus in detail. This review projective guide the integration of new AI technology in the future, too. Overall, this review article could help the beginner who want to introduce the system into field test and who want to get some idea for in the future application.

Author Response

This review addresses the limitations of new technology and suggest the enhancing efficacy detect plant virus in detail. This review projective guide the integration of new AI technology in the future, too. Overall, this review article could help the beginner who want to introduce the system into field test and who want to get some idea for in the future application.

Reply:Thank you for your comment. We have now made revisions based on comments from two other reviewers. Hopefully, the revision still meets your criteria.

Round 2

Reviewer 1 Report

Comments and Suggestions for Authors

The manuscript has been successfully revised and now may be published